# Exploiting an Intermediate Latent Space between Photo and Sketch for Face Photo-Sketch Recognition [note 1]

**DOI:** 10.3390/s22197299

**Published:** 2022-09-26

**Authors:** Seho Bae, Nizam Ud Din, Hyunkyu Park, Juneho Yi

**Affiliations:** 1Department of Electrical and Computer Engineering, Sungkyunkwan University, Suwon 16419, Korea; 2Saudi Scientific Society for Cybersecurity, Riyadh 11543, Saudi Arabia

**Keywords:** face photo-sketch recognition, face photo-sketch synthesis, GAN

## Abstract

The photo-sketch matching problem is challenging because the modality gap between a photo and a sketch is very large. This work features a novel approach to the use of an intermediate latent space between the two modalities that circumvents the problem of modality gap for face photo-sketch recognition. To set up a stable homogenous latent space between a photo and a sketch that is effective for matching, we utilize a bidirectional (photo → sketch and sketch → photo) collaborative synthesis network and equip the latent space with rich representation power. To provide rich representation power, we employ StyleGAN architectures, such as StyleGAN and StyleGAN2. The proposed latent space equipped with rich representation power enables us to conduct accurate matching because we can effectively align the distributions of the two modalities in the latent space. In addition, to resolve the problem of insufficient paired photo/sketch samples for training, we introduce a three-step training scheme. Extensive evaluation on a public composite face sketch database confirms superior performance of the proposed approach compared to existing state-of-the-art methods. The proposed methodology can be employed in matching other modality pairs.

## 1. Introduction

The goal of this work is to find the best matching photos for a given sketch in a face database, especially for software-generated composite sketches. An important application of such systems is to assist law enforcement agencies. In many cases, a face photo of a suspect is unavailable for criminal investigation. Instead, the only clue to identify suspect is a software-generated composite sketch or a hand-drawn forensic sketch based on a description by an eye-witness. Therefore, an automatic method that retrieves the best matching photos from a face database for a given sketch is necessary to quickly and accurately identify a suspect.

Successful photo-sketch matching depends on the solution to how to effectively deal with large modality gap between photo and sketch modalities. Moreover, the insufficiency of sketch samples for training makes photo-sketch recognition an extremely challenging task.

Regarding classical photo-sketch recognition, generative approaches [1,2,3] bring both modalities into a single modality by transforming one of the modalities to the other (either a photo to a sketch, or vice versa) before matching. The main drawback of these methods is their dependency on the quality of the synthetic output, which most of the time suffers due to the large modality gap between photos and sketches. On the other hand, discriminative approaches attempt to extract modality-invariant features or learn a common subspace where both photo and sketch modalities are aligned [4,5,6,7,8,9,10,11,12]. Although these methods formulate photo-sketch recognition through modality-invariant features or a common subspace, their performances are not satisfactory because (1) the distributions of the two modalities are not well aligned in the common feature space and (2) their feature vectors or common spaces fail to provide a rich representation capacity. Recent deep learning-based face photo-sketch recognition methods [6,9,13,14,15,16,17,18,19] perform well compared to classical approaches. However, employing deep learning techniques for face photo-sketch recognition is very challenging because of insufficient training data.

Recently, Col-cGAN [20] proposed a bidirectional face photo-sketch synthesis network. They generated synthetic outputs by using a middle latent domain between photo and sketch modalities. However, their middle latent domain does not provide enough representational power of both modalities. On the other hand, StyleGAN [21] and its updated version, StyleGAN2 [22], produces extremely realistic images by proposing a novel generator architecture. Instead of feeding the input latent code z directly into the generator, they first transform it into an intermediate latent space, W, via a mapping network. This disentangled intermediate latent space, W, offers the generator more control and representational capabilities. Noting the strong representation power of the latent code space of both StyleGAN and StyleGAN2 versions, we take advantage of their architecture in a bidirectional way to set up an intermediate latent space for our photo-sketch recognition problem.

In this paper, we propose a novel method that exploits an intermediate latent space, W, between the photo and sketch modalities, as shown in Figure 1. We employ a bidirectional collaborative synthesis network of the two modalities to set up the intermediate latent space where the distributions of those two modalities are effectively aligned. Additionally, the aforementioned StyleGAN-like architectures enable the intermediate latent space to have strong representational power to successfully match the two modalities. In an earlier version [23] of this work, we only considered the StyleGAN generator. In this work, we extend our work to include StyleGAN2. While StyleGAN applies styles to an image by modulating mean and standard deviation using adaptive instance normalization (AdaIN), StyleGAN2 only modulates standard deviation with weight demodulation. We have experimented on StyleGAN2 in our network and compared it with the performance of StyleGAN.

In Figure 1, the mapping networks, $F_{p}$ and $F_{s}$, learn the intermediate latent codes $w_{p},w_{s}\in W$. To form a homogeneous intermediate space, W, we constrain the intermediate features to be more symmetrical using the $\ell_{1}$ distance between the intermediate latent codes of the photo and the sketch. The intermediate latent space also makes use of feedback from the style generators that conduct translation from photo to sketch and from sketch to photo. This enables the intermediate latent space to have rich representational capacity for both the photo and the sketch that previous methods fail to provide. Once this intermediate latent space is successfully set up, we can then directly take advantage of any state-of-the-art face recognition methods. In our case, we employ AdaCos loss [24].

Moreover, we use a three-step training scheme to resolve the problem of having a very limited number of training sketch samples. In the first step, we only learn image-to-image translation without AdaCos on paired photo-sketch samples. This serves the purpose of learning an initial intermediate latent space. Then, in the second step, we pre-train the photo mapping network, $F_{p}$, only with AdaCos, using a publicly available large photo dataset. This helps our model overcome the problem of insufficient sketch samples to train our deep network robustly for the target task. Lastly, we fine-tune the full network on a target photo/sketch dataset. More details of the model training are discussed in Section 3.

The main contributions of our work are summarized as follows.

We propose a novel method for photo-sketch matching that exploits an intermediate latent space between the photo and sketch modalities;
-We set up the intermediate latent space through a bidirectional collaborative synthesis network;-This latent space has rich representational power for photo/sketch recognition by employing StyleGAN-like architectures such as StyleGAN or StyleGAN2;A three-step training scheme helps overcome the problem of insufficient sketch training samples;Extensive evaluation on challenging publicly available composite face sketch databases shows the superior performance of our method compared with state-of-art methods.

The source code is available at https://github.com/seho-bae/face-photo-sketch-recognition-bidirectional-network (accessed on 12 July 2022). The rest of the paper is organized as follows. Section 2 states related works. We elaborate details of our method in Section 3. Section 4 shows experimental results.

## 2. Related Work

The face photo-sketch recognition problem has been extensively studied in recent years. Researchers have studied sketch recognition for various face sketch categories such as hand-drawn viewed sketches, hand-drawn semi-forensic sketches, hand-drawn forensic sketches, and software-generated composite sketches. Compared to hand-drawn viewed sketches, other sketch categories have much larger modality gaps due to the errors that come from forgetting (semi-forensic/forensic), understanding the description (forensic), or limitations of the components of the software (composite). Recent studies focus on more challenging composite and forensic sketches.

Trivial recognition methods can be categorized into generative and discriminative approaches.

Generative methods convert images from one modality into another modality, usually from a sketch to a photo, before matching. Then, a simple homogeneous face recognition method can be used for matching. Various techniques have been utilized for synthesis, such as a Markov random field model [1], local linear embeding (LLE) [2], and multi-task Gaussian process regression [3]. However, the recognition performance of these approaches is strongly reliant on their synthesis results, which most of the time suffer due to the large modality gap between the two modalities.

Discriminative methods attempt to learn a common subspace or extract particular features in order to reduce the modality gap between photos and sketches of the same identity while increasing the gap between different identities. Representative methods in this category include partial linear squares (PLS) [4,5], coupled information-theoretic projection (CITP) [6], local feature-based discriminant analysis (LFDA) [7], canonical correlation analysis (CCA) [8], and self-similarity descriptor (SSD) dictionary [9]. Han et al. [10] computed the similarity between a photo and composite sketch using a component-based representation technique. Multi-scale circular Weber’s local descriptor (MCWLD) was utilized in Bhatt et al. [11] to solve the semi-forensic and forensic sketch recognition problem. Using graphical representation-based heterogeneous face recognition (G-HFR) [12], the authors graphically represented heterogeneous image patches by employing Markov networks and designed a similarity metric for matching. These methods fail when the learned feature/common subspace do not have enough representational capacity for both photo and sketch modalities. In contrast, our method projects photos and sketches on a homogeneous intermediate space, where the distribution of the two modalities is better aligned with rich representational power.

Over the past few years, deep learning-based algorithms have been developed for face photo-sketch recognition [6,9,13,14,15,16,17,18]. Kazemi et al. [13] and Iranmanesh et al. [14] proposed attribute-guided approaches by introducing attribute-centered loss function and a joint loss function of identity and facial attribute classification, respectively. Liu et al. designed an end-to-end recognition network using a coupled attribute guided triplet loss (CAGTL). It plausibly eliminates defects of incorrectly estimated attributes [15] during training. Iterative local re-ranking with attribute guided synthesis based on GAN was introduced in [16]. Peng et al. proposed DLFace [17], which is a local descriptor approach based on deep metric learning, while in [18], a hybrid feature model was employed by fusing traditional HOG features with deep features. The largest obstacle to utilizing deep learning techniques for face photo-sketch recognition is the scarcity of sketch data. Even the largest publicly viewed sketch database [6] has only 1194 pairs of sketches and photos, and the composite sketch database [9] has photos and sketches of 123 identities. To overcome this problem, most approaches employ relatively shallow networks, data augmentation, or pre-training on a large-scale face photo database.

Recently, cosine-based softmax losses [24,25,26,27] have achieved great success in face photo recognition. SphereFace [25] applied a multiplicative penalty to the angles between the deep features and their corresponding weights. Follow-up studies improved the performance by changing the penalising measure to an additive margin in cosine [26] and angle [27]. AdaCos [24] outperforms previous cosine-based softmax losses by leveraging an adaptive scale parameter to automatically strengthen the supervision during training. However, the direct application of these methods to photo-sketch recognition is not satisfactory because they have not properly dealt with the modality gap.

## 3. Proposed Method

Our proposed framework takes advantage of a bidirectional photo/sketch synthesis network to set up an intermediate latent space as an effective homogeneous space for face photo-sketch recognition. The mutual interaction of the two opposite synthesis mappings occurs in the bidirectional collaborative synthesis network. The complete structure of our network is illustrated in Figure 2a. Our network consists of mapping networks $F_{p}$ and $F_{s}$, style generators $G_{p}$ and $G_{s}$, and discriminators $D_{p}$ and $D_{s}$. $F_{p}$ and $F_{s}$ share their weights.

The mapping networks, $F_{p}$ and $F_{s}$, learn to encode photo and sketch images into their respective intermediate latent codes, $w_{p}$ and $w_{s}$. Then, $w_{p}$ and $w_{s}$ are fed into the two opposite style generators $G_{s}$ and $G_{p}$ to perform photo-to-sketch and sketch-to-photo mapping, respectively. We employ a StyleGAN-like architecture to make the intermediate latent space be equipped with rich representational power. We also introduce a loss function to regularize the intermediate latent codes of two modalities, enabling them to learn a same feature distribution. Through this strategy, we learn a homogeneous intermediate feature space, W, that shares common information of the two modalities, thus producing the best results for heterogeneous face recognition. To enforce latent codes in W separable in the angular space, we learn AdaCos [24] for the photo-sketch recognition task.

$F_{p}$ and $F_{s}$ employ a simple encoder architecture that contains convolution, max pooling and fully connected layers. The style generators, $G_{p}$ and $G_{s}$, consist of several style blocks and transposed convolution layers as in [21]. StyleGAN [21] and StyleGAN2 [22] use different style block architectures, as shown in Figure 2b. The details are described in Section 3.1. The discriminators, $D_{p}$ and $D_{s}$, distinguish generated photo/sketch and real samples by taking corresponding concatenated photo and sketch. We use a PatchGAN architecture [28] of 70 × 70. Unlike the discriminator in [20], our discriminator uses instance normalization instead of batch normalization.

### 3.1. StyleGAN and StyleGAN2 Generators

Both StyleGAN [21] and StyleGAN2 [22] transfer learned constant input to an output image by adjusting the style of an image at each style block using the latent code, w, as illustrated in Figure 2. The difference between the two versions is in the internal structure of the style blocks.

The style block of our StyleGAN based ganerator is depicted on the left of Figure 2b. The latent code defining a style is applied to an image through adaptive instance normalization (AdaIN). AdaIN first normalizes the feature maps, then modulates mean and standard deviation with the latent code. Unlike [21], we do not use noise inputs and progressively growing architecture because the sole purpose of our style generators is to help the homogeneous intermediate latent space retain common representational information of the two modalities for reducing the modality gap between them. Our style generators are very light as compared to that of [21] due to the limited number of training samples.

The StyleGAN2 version of our style block is shown on the right of Figure 2b. StyleGAN2 [22] replaces AdaIN with weight modulation to remove blob-shaped artifacts in the original StyleGAN. Weight demodulation, first, scales the weights of the convolution layer using latent code to modulate the standard deviation of the convolution outputs (mod block in Figure 2b). Then, the outputs are scaled by the $\ell_{2}$ norm of the corresponding weights before the next style block. This scaling is baked into the convolution weights directly (demod block in Figure 2b). The crucial difference from the StyleGAN’s style block is that StyleGAN2 modulates standard deviation only. In StyleGAN2, mean of the feature maps is changed by bias after modulation.

Furthermore, ref. [22] also proposes a skip connection-based generator and residual network discriminator with path length regularization in order to synthesize very clear high-resolution images (1024 × 1024). However, we do not use these techniques because our target image resolution is low (256 × 256), and the number of training data is very small.

Both versions of the generator improve the recognition performance by providing more representation capacity to the intermediate latent space. We compare the StyleGAN and StyleGAN2 versions of our network in Section 4.3.

### 3.2. Loss Functions

To learn the identity recognition, we use AdaCos loss [24], $L_{AdaCos}$, which measures the angular distance in the W space. It minimizes the distance of intra-class features while maximizing the distance of inter-class features. $L_{AdaCos}$ is defined as
(1)$$\begin{array}{ccc} L_{AdaCos}=-\frac{1}{N}\sum_{i=1}^{N}\log\frac{e^{s\cdot \cos\theta_{i,y_{i}}}}{e^{s\cdot \cos\theta_{i,y_{i}}}+B_{i}}, & \; & B_{i}=\sum_{k\neq y_{i}}e^{s\cdot \cos\theta_{i,y_{i}}}\approx C-1, \end{array}$$
where $y_{i}$ is the corresponding label of the *i*-th face image in the current mini-batch with size *N*, and *C* is the number of classes. *s* is the adaptive scaling parameter. At the *t*-th iteration of the training process, the scaling parameter is formulated as
(2)$$s^{\left(t\right)}=\left\{\begin{array}{cc} \sqrt{2}\cdot \log\left(C-1\right) & t=0\\ \frac{\log B_{\mathrm{avg}\;}^{\left(t\right)}}{\cos\left(\min\left(\frac{\pi }{4},\theta_{\mathrm{med}\;}^{\left(t\right)}\right)\right)} & t\geq 1, \end{array}\right.$$ where $\theta_{med}^{\left(t\right)}$ is the median value of $\theta_{i,y_{i}}$ in the current mini-batch.

To train the bidirectional photo/sketch synthesis part of the whole network, we use the GAN loss function, $L_{GAN}$ [29], along with the similarity loss, $L_{s}$, as follows.
(3)$$\begin{array}{c} L_{GAN}=\min_{G}\max_{D}E\left[\log D_{s}\left(x_{s}\right)\right]+E\left[1-\log D_{s}\left(G_{s}\left(x_{p}\right)\right)\right]\\ +E\left[\log D_{p}\left(x_{p}\right)\right]+E\left[1-\log D_{p}\left(G_{p}\left(x_{s}\right)\right)\right], \end{array}$$
(4)$$L_{s}=E[|x_{s}-y_{p\to s}\left|+\right|x_{p}-y_{s\to p}\left|\right].$$

$L_{GAN}$ helps to generate real and natural-looking synthetic outputs, while the similarity loss, $L_{s}$, measures the pixel-wise $\ell_{1}$ distance between generated and real photo/sketch images.

In addition, to regularize and enforce the same distribution for a photo, $w_{p}$, and a sketch, $w_{s}$, in the intermediate latent space, we introduce a collaborative loss,
(5)$$\begin{array}{c} L_{w}=E[|w_{p}-w_{s}\left|\right]. \end{array}$$

It minimizes the $\ell_{1}$ distance between $w_{p}$ and $w_{s}$ of the same identity.

Thus, the joint loss function used to train our network is as follows:(6)$$\begin{array}{c} L=L_{AdaCos}+\lambda_{GAN}L_{GAN}+\lambda_{s}L_{s}+\lambda_{w}L_{w}. \end{array}$$

$\lambda_{GAN}$, $\lambda_{s}$, and $\lambda_{w}$ in Equation (Equation 6) control the relative importance of each loss function in the bidirectional photo/sketch synthesis task. We used $\lambda_{GAN}$ = 1, $\lambda_{s}$ = 10, and $\lambda_{w}$ = 1 in our experiments.

### 3.3. Training

To overcome the problem of an insufficient amount of paired photo/sketch training data, we introduce a simple and effective three-step training scheme, as shown in Figure 3. In step 1, we train the bidirectional photo/sketch synthesis network using paired photo-sketch training samples to set up an initial homogeneous intermediate latent space, W. We use our joint loss function in Equation (Equation 6), excluding the AdaCos loss function, $L_{AdaCos}$. In step 2, we pre-train the photo mapping network, $F_{p}$, using AdaCos loss only on the publicly available large photo database CelebA [30] to overcome the problem of insufficient sketch training samples. Then, we train our full network in step 3 using the whole joint loss function in Equation (Equation 6) on target photo/sketch samples.

## 4. Experiments

The recognition accuracies of our network presented in this section are average results over five experiments with random partitions.

### 4.1. Data Description and Implementation Details

We have conducted our experiments using the e-PRIP composite sketch dataset [9]. The e-PRIP dataset consists of four different composite sketch sets of 123 identities. However, only two of them are publicly available: the composite sketches created by an Indian user adopting the FACES tool [31] and an Asian artist using the Identi-Kit tool [32]. We have used 48 identities for training and the remaining 75 identities for testing.

To mimic a real law-enforcement scenario where multiple numbers of suspects are selected from a large photo database, we have performed the experiments with extended galleries. As some photos in the extended galleries of the previous works [12,13,14,15,17] are not publicly available, we have mimicked their galleries as closely as possible using the photos from the other publicly available databases for fair comparison. Two galleries of different sizes were used in the previous studies. Following [13,14], we have constructed an extended gallery of 1500 subjects including probe images by using photos from ColorFERET [33], the Multiple Encounter Dataset (MEDS) [34], CUFS [35], and the IIIT-D sketch database [11]. To compare the performance with [12,17] and [15], we have built another extended gallery of 10,075 subjects using face photos collected from the aforementioned photo databases and the faces in the wild-a (LFW-a) [36] and Flickr-Faces-HQ (FFHQ) [21] datasets.

We aligned all photos and sketches by eye position and initially cropped them to 272 × 272. Then, they were randomly cropped to 256 × 256 during training. We optimized our network using the Adam optimizer with a learning rate of 0.0002 and a batch size of 8 in steps 1 and 3 of training. We used a learning rate of 0.0005 and a batch size of 32 in step 2. We trained our networks for 3000 epochs on the CUFS [35] viewed sketch database in step 1 of training, for 50 epochs on CelebA [30] in step 2, and for 3000 epochs on the target database in step 3.

### 4.2. Photo-Sketch Recognition Results

In this section, we compare the performance of our method with that of representative state-of-the-art photo-sketch matching methods on the two subsets of the e-PRIP dataset [9]. Let us denote them as FACES (In) and Identikit (As), respectively.

In a real scenario, when matching forensic sketches with large galleries, law enforcement officers consider the top *R* retrieval results as potential suspects [7]. Thus, we have compared rank 50 accuracy, which is the most commonly used criteria.

The results with a gallery with a size of 1500 are presented in Table 1, where the accuracies for Kazemi et al. and Iranmanesh et al. are obtained from their CMC curves. Both versions of our method outperformed the SOTA. In particular, the StyleGAN version achieved 93.86% rank 50 accuracy on Faces (In), which was 13.86% higher than [14], and 90.40% on Identikit (As), which was 7.4% higher than [14]. Table 2 shows the comparison results of our method with the previous state-of-the-art representative methods with a gallery of size 10,075. As can be seen in the table, our StyleGAN version shows far better performance of 92.78% and 88.26% rank 50 accuracies on Faces (In) and Identikit (As), respectively, with large margins. Our StyleGAN2 version achieved 90.14% and 80.28% rank 50 accuracies on Faces (In) and Identikit (As), respectively, while the previous SOTA method CAGTL [15] yielded 78.13% and 67.20%. Although StyleGAN outperforms StyleGAN2, the performance of StyleGAN2 version is still much higher than that of SOTA. These results show that our bidirectional collaborative synthesis network based on StyleGAN or StyleGAN2 learns an effective intermediate latent space with rich representational power for the face photo-sketch recognition task.

Figure 4 shows the CMC curves of our method for StyleGAN and StyleGAN2. They indicate the strength in the top *R* retrieval of both versions.

### 4.3. Effect of Generator Architecture

In this section, to verify the power of the StyleGAN-like generators in our networks, we compare our models of StyleGAN and StyleGAN2 generators with a traditional U-net [37]-based generator. We designed a U-net version model which has an encoder of seven convolution layers with a fully connected layer and a decoder of eight transposed convolution layers. The decoder layers receive the information of encoder layers through skip connections. The latent code is of a 512-dimensional vector, the same as for the StyleGAN and StyleGAN2 versions.

The results in Table 3 show the latent spaces from StyleGAN and StyleGAN2 for recognition. Due to the representation capacity provided by the StyleGAN-like architecture, our models achieved much higher rank 50 accuracies than the traditional U-net based generator, especially for the Identikit (As) dataset. The StyleGAN version network outperforms the StyleGAN2 version with less margin compared to the U-net version. In StyleGAN2, learned biases replace the mean modulation of StyleGAN. We think that the performance gap between StyleGAN and StyleGAN2 models comes from these biases, which are not perfectly trained. Though our three-step training scheme alleviates the data size problem, 48 training samples are still insufficient to fully train the whole synthesis networks. Thus, learned biases fail to replace the mean modulation, and thereby, StyleGAN2 underperforms the StyleGAN version network.

The original StyleGAN2 work [22] also proposed a generator and discriminator design for high-resolution image synthesis. We implemented these network designs in our StyleGAN2 version network and compared the results in Table 3. As expected, the skip connection-based generator and residual network-based discriminator design is not helpful for our face photo-sketch recognition. The recognition performance degraded as our network became heavier in the architecture design. This is because our target image resolution is very low (256 × 256) compared to the target resolution of 1024 × 1024, and the number of training data is insufficient. We opt to not apply these architecture designs to our StyleGAN2 version network.

### 4.4. Effect of Bidirectional Collaborative Synthesis of Photo-to-Sketch and Sketch-to-Photo

To show the effectiveness of our bidirectional collaborative synthesis network approach on the recognition task, we provide a comparison with three different variants from the full network. In the first variant, we removed the style generators, $G_{s}$ and $G_{p}$, from the network in Figure 2 and trained the mapping networks, $F_{p}$ and $F_{s}$, using the AdaCos loss function. That is, the first variant could not take any advantage of the synthesis network. For this variant, the mapping networks were pre-trained for 50 epochs on the CelebA photo database [30], then fine-tuned for 3000 more epochs on the target database. For the second and third variants, we trained a unidirectional synthesis-based photo-sketch recognition network by using only one of the style generators, either $G_{s}$ or $G_{p}$. These two variants employed the three-step training scheme as in the full network. We have tested these variants for both StyleGAN and StyleGAN2 versions.

The comparison results in Table 4 indicate that the addition of either a photo or sketch generator improves the recognition accuracy. For both StyleGAN and StyleGAN2 versions, the unidirectional sketch-to-photo network shows better performance than the unidirectional photo-to-sketch network. This is because the sketch-to-photo network translates the information-poor input to information-rich output, thus providing better representational feedback to the intermediate latent space as compared to the photo-to-sketch network. For unidirectional synthesis variants, StyleGAN2 achieved higher rank 50 accuracies than StyleGAN. However, it still cannot provide enough representational power. Our full network, which exploits the bidirectional collaborative synthesis network, dramatically improved the recognition performance. This is because our bidirectional synthesis network warrants the intermediate latent space to have important representational information by utilizing the mutual interaction between the two opposite mappings.

Figure 5 shows examples of synthesis results generated by our networks. The results show that bidirectional networks produce better synthesis results than the unidirectional networks regardless of style generator architecture. Utilizing the mutual interaction between opposite mappings helps make the intermediate latent space have important representational information.

### 4.5. Effect of Three-Step Training Scheme

To validate the effectiveness of the proposed three-step training scheme, we compared three different training settings in Table 5. For this, we trained our StyleGAN and StyleGAN2 version models (1) using only step 3, that is, without pre-training, (2) using step 2 and step 3, and (3) using all the three steps.

We can see that there is significant improvement in recognition accuracy when using pre-training (step 2) which enhances our encoders, especially for the Faces (In) dataset. This shows the power of large-scale pre-training in solving the data scarcity problem. The combination of all the three training steps further boosts the recognition performance. Step 1 provides an effective initialization of the intermediate latent space between a photo and a sketch for large-scale training in step 2. As the last row in Table 5 shows, our three-step training strategy effectively overcomes the problem of insufficient sketch training samples. Compared to the StyleGAN version, the StyleGAN2 version network takes greater benefit from the three-step training scheme. We can see that without pretraining, it is not even possible to train the StyleGAN2 version network (Table 5). This shows that the StyleGAN2 version requires more training data than the StyleGAN version.

### 4.6. Collaborative Loss, $L_{w}$

In this section, we analyze the effect of collaborative loss, $L_{w}$, on the recognition accuracy. We experimented with both the StyleGAN and StyleGAN2 versions of our networks as we changed the value of $\lambda_{w}$. Table 6 shows the results for different values of $\lambda_{w}$ on the extended gallery setting of 1500 samples.

The performance of $\lambda_{w}=0$ is poor. $\lambda_{w}=0$ means that our network is not using collaborative loss $L_{w}$. The network is unable to constrain the two mappings symmetrical. The accuracy significantly improves when network use collaborative loss $L_{w}$ for both StyleGAN and StyleGAN2 versions. Through many experiments, we have found that $\lambda_{w}$ = 1 produces the best result for our task. These results show that our collaborative loss helps regularize the intermediate latent representations of the two different modalities, effectively aligning the two modalities in the intermediate latent space. However, as $\lambda_{w}$ gets too large, the performance degrades, as can be seen in Table 6. We think that a large value of $\lambda_{w}$ places too much emphasis on making latent codes symmetrical and breaks the learning balance of the latent space between representational capacity and symmetrical mapping.

Figure 6 shows examples of synthesis results produced by our style generators for different values of $\lambda_{w}$. There is a general trend that better synthesis results yield better recognition accuracies. For $\lambda_{w}$ = 10, the results collapsed to the same synthesis result for most of the target samples. In particular, the output images of the StyleGAN2 version collapsed from very early epochs, resulting in significant degradation of recognition performance. This shows that too much weightage to the collaborative loss strongly enforces the same latent distribution, while the representational capacity of the latent space is relatively ignored.

### 4.7. Similarity Loss, $L_{s}$

Figure 7 shows the results produced by our StyleGAN and StyleGAN2 generators for three simple variations of $L_{s}$. First, we used pixel-wise $\ell_{1}$ distance only as our $L_{s}$. Second, we used only patch-wise structural similarity (SSIM) loss [38]. Third, we employed SSIM loss along with $\ell_{1}$ distance for $L_{s}$. Figure 7 shows that using only SSIM loss for $L_{s}$ produces the worst synthetic results, yielding the lowest recognition accuracy for both StyleGAN and StyleGAN2 versions, as can be seen in Table 7. On the other hand, $\ell_{1}$ produces the best recognition results compared to the other two settings. Our observation is that SSIM loss provides extra structural information for synthesis, but it does not help for recognition. Thus, we opt to use only $\ell_{1}$ distance as our $L_{s}$ in the joint loss function in Equation (Equation 6).

## 5. Conclusions

We have proposed a novel deep learning-based face photo-sketch recognition method by exploiting a homogeneous intermediate latent space between photo and sketch modalities. For this, we have explored a bidirectional photo/sketch synthesis network based on a StyleGAN-like architecture in two versions: StyleGAN and StyleGAN2. In addition, we employ a simple three-step training scheme to overcome the problem of insufficient photo/sketch data. The experiment results have verified the effectiveness of our method.

## Figures and Tables

**Figure 1 sensors-22-07299-f001:**
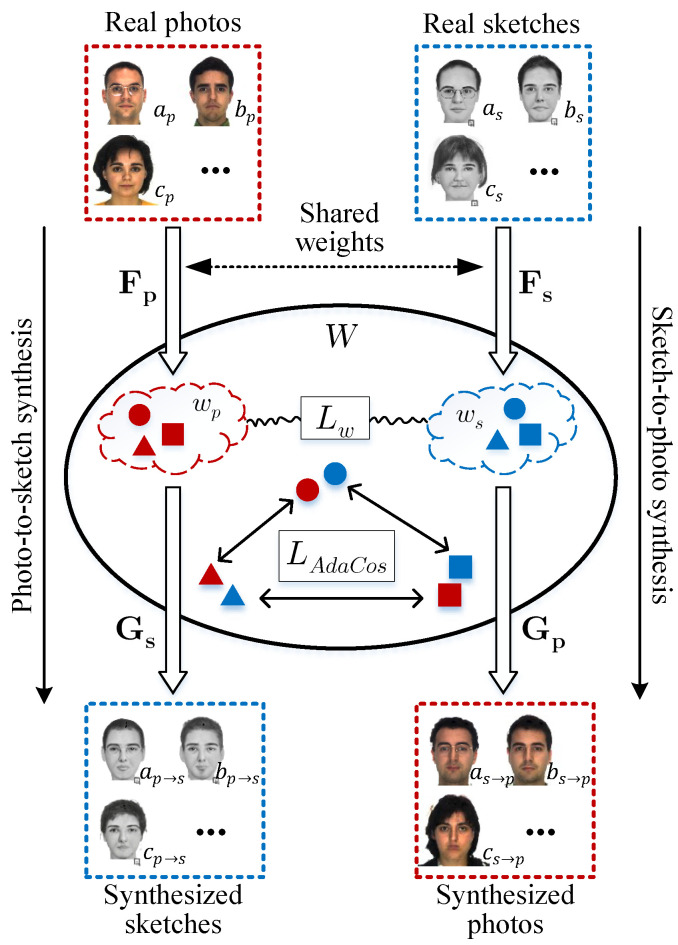
Our proposed framework takes advantage of a bidirectional photo/sketch synthesis network to set up an intermediate latent space as an effective homogeneous space for face photo-sketch recognition. We employ StyleGAN-like architectures such as StyleGAN and StyleGAN2 to make the intermediate latent space be equipped with rich representational power. The mapping networks, $F_{p}$ and $F_{s}$, learn to encode photo and sketch images into their respective intermediate latent codes, $w_{p}$ and $w_{s}$. We learn AdaCos [24] to enforce the separability of latent codes of different identities in the angular space for the photo-sketch recognition task.

**Figure 2 sensors-22-07299-f002:**
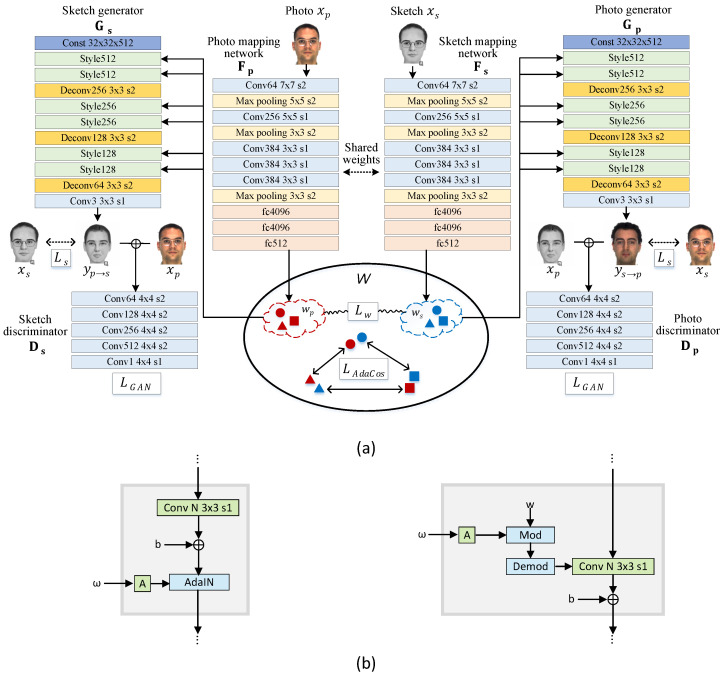
(**a**) The overall architecture of the proposed network. Mapping networks, $F_{p}$ and $F_{s}$, map photo and sketch images to intermediate latent codes $w_{p}$ and $w_{s}$. These latent codes are then fed into the two opposite style generators $G_{s}$ and $G_{p}$. $G_{s}$ generates a sketch from a photo, $y_{p\to s}$, while $G_{s}$ generates a photo from a sketch, $y_{s\to p}$. The collaborative loss $L_{w}$, which is $\ell_{1}$ distance between $w_{p}$ and $w_{s}$ of a same identity, constrains the intermediate modality features to be more symmetrical. Through this strategy, we learn an intermediate latent space, W, that retains the common and representational information of the photo and sketch. We apply AdaCos loss, $L_{AdaCos}$, to the intermediate latent space, W, directly to perform photo-sketch recognition by comparing the cosine distance between intermediate latent features, $w_{p}$ and $w_{s}$. (**b**) The internal structures of style blocks in our networks: StyleGAN (**left**) and StyleGAN2 (**right**). Symbol A is an affine transformation that tranfers latent code to the style form. StyleGAN2 replaces AdaIN with weight demodulation. Unlike [21,22], we do not use noise input because it does not serve for recognition.

**Figure 3 sensors-22-07299-f003:**
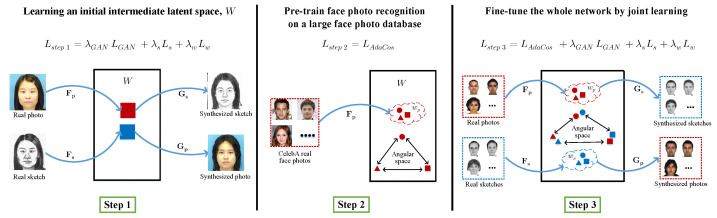
A three-step training scheme to overcome the problem of an insufficient amount of paired photo-sketch training samples. We employ three-step training. Step 1: Pre-train the bidirectional photo/sketch synthesis network to learn an initial intermediate latent space, W, between photo and sketch. Step 2: Pre-train the photo mapping network, $F_{p}$, on a large face photo database. Step 3: Fine-tune the whole network on the target photo/sketch database.

**Figure 4 sensors-22-07299-f004:**
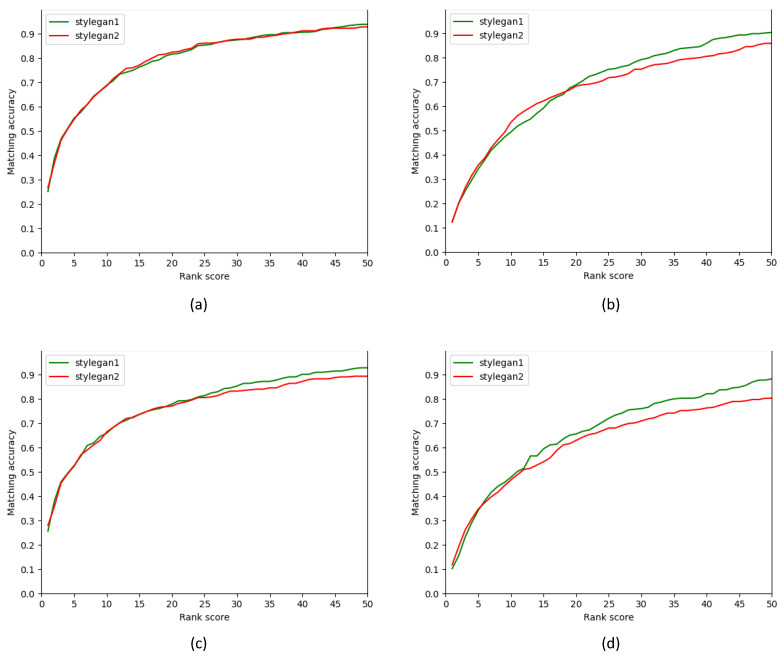
CMC curves of our StyleGAN and StyleGAN2 version models on different databases. (**a**) Faces (IN) with gallery size 1500, (**b**) IdentiKit (As) with gallery size 1500, (**c**) Faces (IN) with gallery size 10,075, and (**d**) IdentiKit (As) with gallery size 10,075.

**Figure 5 sensors-22-07299-f005:**
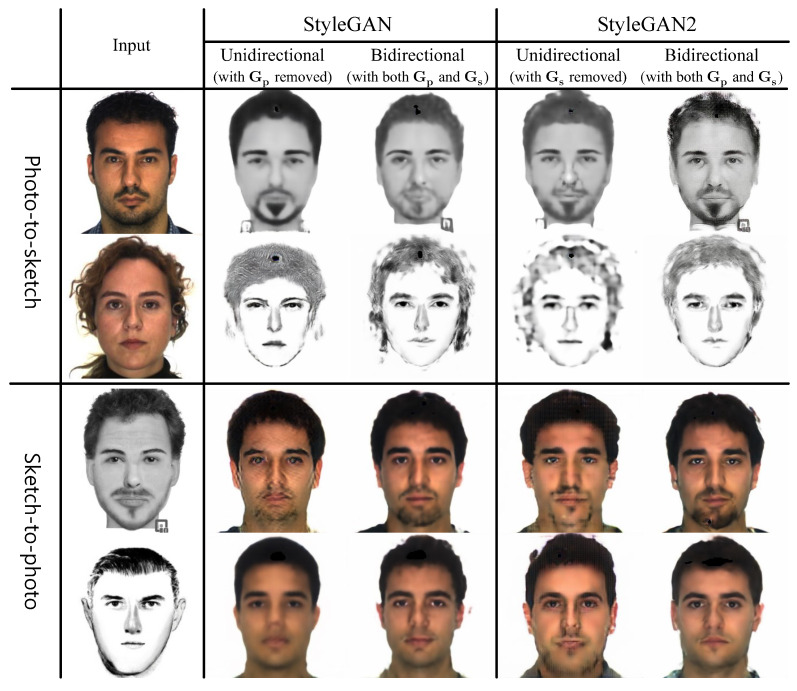
Synthesis results of unidirectional and bidirectional collaborative synthesis networks with StyleGAN/StyleGAN2-based generators. The first and third rows are for Faces (In), and the second and fourth rows are for Identikit (As). For both StyleGAN and StyleGAN versions, bidirectional networks produce better synthetic images as compared to unidirectional networks (please view in color).

**Figure 6 sensors-22-07299-f006:**
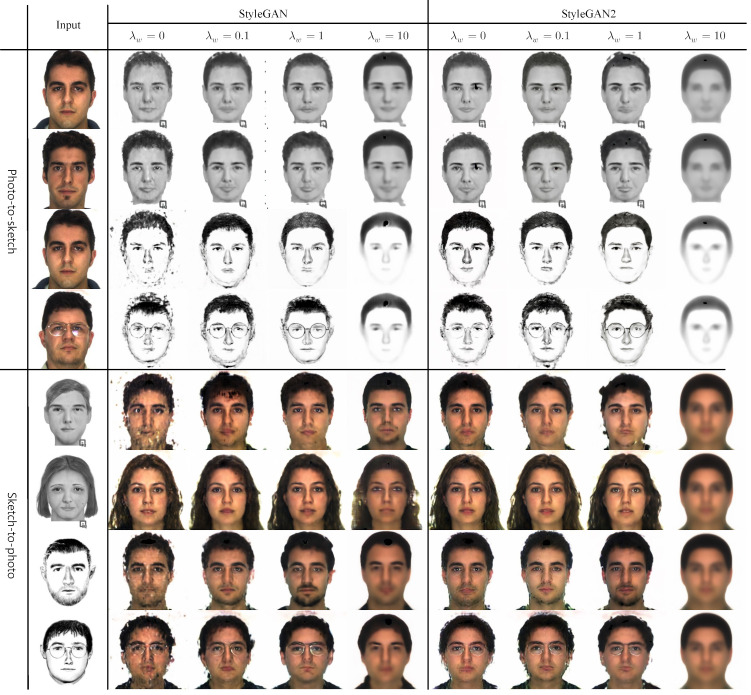
Synthesis results of our style generators for different values of $\lambda_{w}$. In both photo-to-sketch and sketch-to-photo synthesis, first and second rows are for Faces (In), while third and fourth rows are for Identikit (As). Images collapse with high $\lambda_{w}$ so that the network could not learn representational information of the photo and sketch. $\lambda_{w}$=1 shows the best synthesis results (please view in color).

**Figure 7 sensors-22-07299-f007:**
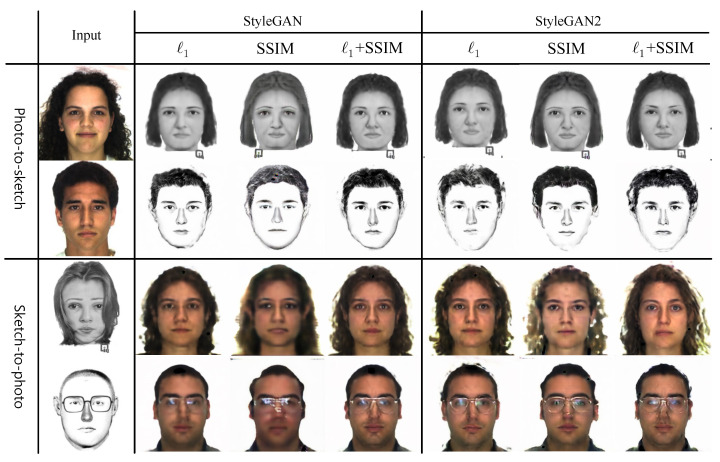
Synthesis results of our StyleGAN and StyleGAN2 generators for three different versions of $L_{s}$. First and third rows are trained on Faces (In), while second and fourth rows are trained on Identikit (As) (please view in color).

**Table 1 sensors-22-07299-t001:** Rank 50 recognition accuracy (%) on the e-PRIP database with a gallery size of 1500.

Method	Faces (In)	Identikit (As)
Kazemi et al. [13]	77.50	81.50
Iranmanesh et al. [14]	80.00	83.00
Ours: StyleGAN	**93.86**	**90.40**
Ours: StyleGAN2	92.80	85.84

**Table 2 sensors-22-07299-t002:** Rank 50 recognition accuracy (%) on the e-PRIP dataset with a gallery size of 10,075.

Method	Faces (In)	Identikit (As)
G-HFR [12]	-	51.22
DLFace [17]	70.00	58.93
CAGTL [15]	78.13	67.20
Ours: StyleGAN	**92.78**	**88.26**
Ours: StyleGAN2	90.14	80.28

**Table 3 sensors-22-07299-t003:** Rank 50 recognition accuracy (%) on the e-PRIP dataset with a gallery size of 1500 for different generator architectures.

Generator Architecture	Faces (In)	Identikit (As)
U-Net	83.2	54.14
StyleGAN	**93.86**	**90.40**
StyleGAN2	92.80	85.84
skip G and residual D	84.26	81.84

**Table 4 sensors-22-07299-t004:** Rank 50 recognition accuracy (%) on the e-PRIP dataset with a gallery size of 1500 for the synthesis networks.

Generator	Method	Faces (In)	Identikit (As)
None	Only mapping networks	19.74	43.72
StyleGAN	Photo-to-sketch (with $G_{p}$ removed)	68.54	61.58
	Sketch-to-photo (with $G_{s}$ removed)	73.84	73.88
	Our full network (with both $G_{p}$ and $G_{s}$)	**93.86**	**90.40**
StyleGAN2	Photo-to-sketch (with $G_{p}$ removed)	87.48	79.22
	Sketch-to-photo (with $G_{s}$ removed)	88.52	82.40
	Our full network (with both $G_{p}$ and $G_{s}$)	92.8	85.84

**Table 5 sensors-22-07299-t005:** Rank 50 recognition accuracy (%) on the e-PRIP dataset with a gallery size of 1500 for training scheme.

Generator	Method	Faces (In)	Identikit (As)
StyleGAN	Without pre-training (step 3 only)	25.32	46.14
	Two-step training (step 2 + step 3)	90.66	89.60
	Three-step training (step 1 + step 2 + step 3)	**93.86**	**90.40**
StyleGAN2	Without pre-training (step 3 only)	5.34	20.00
	Two-step training (step 2 + step 3)	89.6	81.6
	Three-step training (step 1 + step 2 + step 3)	92.80	85.84

**Table 6 sensors-22-07299-t006:** Rank 50 recognition accuracy (%) on the e-PRIP dataset with a gallery size of 1500 for $\lambda_{w}$.

Method	StyleGAN		StyleGAN2	
	Faces (In)	Identikit (As)	Faces (In)	Identikit (As)
$\lambda_{w}$ = 0	72.00	66.40	72.26	63.98
$\lambda_{w}$ = 0.1	89.32	82.68	91.46	81.60
$\lambda_{w}$ = 0.5	89.60	85.60	89.34	76.52
$\lambda_{w}$ = 1	**93.86**	**90.40**	92.80	85.84
$\lambda_{w}$ = 5	85.34	84.28	65.06	50.96
$\lambda_{w}$ = 10	83.72	83.74	22.42	22.92

**Table 7 sensors-22-07299-t007:** Rank 50 recognition accuracy (%) on the e-PRIP dataset with a gallery size of 1500 for $L_{s}$.

Method		Faces (In)	Identikit (As)
StyleGAN	$\ell_{1}$	**93.86**	**90.40**
	SSIM	81.86	79.74
	$\ell_{1}$ + SSIM	91.98	89.34
StyleGAN2	$\ell_{1}$	92.80	85.84
	SSIM	88.00	80.80
	$\ell_{1}$ + SSIM	89.88	82.38

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
