# Peer review of "Exploiting an Intermediate Latent Space between Photo and Sketch for Face Photo-Sketch Recognition"

_sensors, 2022, doi:10.3390/s22197299_

Round 1

Reviewer 1 Report

 The manuscript used an intermediate latent space between photo and sketch for face photo-sketch recognition. In general, the manuscript gives a good description of method. However, I believe the manuscript needs to be revised.

 1. It is necessary to explain clearly the superiority of the proposed method through comparison with other methods.

2. There is no experimental comparison of the algorithm with previously known work.

Reviewer 2 Report

The work provides a method for photo-sketch recognition. According to the authors the method works better than other published methods where they compare it with. The authors used the public part of a database and used part of the images as training set and part as test set, as is usual in this kind of research. Though it is not very reproducible, since it depends on which selection is made as such. Also a question is how many images the other researchers used and if the comparison of results can be made this way. So I would like to ask the authors to clarify this issue. 

Round 2

Reviewer 2 Report

The paper has been improved and is good for publication